# Influence of Processing Parameters on the Conduct of Electrical Resistance Sintering of Iron Powders

**Fátima Ternero [1],\***, **Raquel Astacio [1]**, **Eduardo S. Caballero [1]**, **Francisco G. Cuevas [2]** and **Juan M. Montes [1]**

1   Engineering of Advanced Materials Group, Higher Technical School of Engineering, University of Seville. Camino de los Descubrimientos s/n, 41092 Sevilla, Spain; rastacio@us.es (R.A.); esanchez3@us.es (E.S.C.); jmontes@us.es (J.M.M.)

2   Department of Chemical Engineering, Physical Chemistry and Materials Science, Higher Technical School of Engineering, University of Huelva, Campus El Carmen, Avda. 3 de marzo s/n, 21071 Huelva, Spain; fgcuevas@dqcm.uhu.es

\*   Correspondence: fternero@us.es; Tel.: +34-954-487305

**Abstract:** The influence of the applied pressure and electrical parameters on the macrostructure of specimens consolidated by the medium-frequency electrical resistance sintering technique (MF-ERS) is analysed in this work. This technique is based on the application of pressure to a mass of conductive powder that, simultaneously, is being crossed by a high intensity and low voltage electric current. The simultaneous action of the pressure and the heat released by the Joule effect causes the densification and consolidation of the powder mass in a very short time. The effect of the current intensity and heating time on the global porosity, the porosity distribution, and the microhardness of sintered compacts is studied for two applied pressures (100 and 150 MPa). For the different experiments of electrical consolidation, a commercially available pure iron powder was chosen. For comparison purposes, the properties of the compacts consolidated by MF-ERS are compared with the results obtained by the conventional powder metallurgy route (cold pressing and furnace sintering). Results show that, as expected, higher current intensities and dwelling times, as well as higher pressures and the consolidation of compacts with lower aspects ratios, produce denser materials.

**Keywords:** electrical resistance sintering; electrical consolidation; MF-ERS; FAST; powder metallurgy; hot pressing; iron powder

## 1. Introduction

The use of electricity as a direct method for sintering metallic and nonmetallic powders was suggested many times in the 20th century and is still a topic of extraordinary interest. The first known patent for electric current sintering was registered by Lux [1], in 1906. Some years later came the studies by Taylor [2] (in 1933) and Lenel [3] (in the 1950s), who called this technique electrical resistance sintering under pressure (ERS). Later, during the 70s and 80s, and mainly by Soviet and Japanese researchers, the ERS technique received a new impulse [4–11]. At present, many international researchers study different modalities of field assisted sintering techniques (FAST), which is the common name used for these powder metallurgy (PM) electrical consolidation techniques.

During this long period, many FAST variants have been developed, with the general final objective of their use on the industrial scale (see Grasso et al. [12], Orrù et al. [13], and Olevsky et al. [14]). Perhaps the most popular technique is the so-called spark plasma sintering (SPS), in which low wear resistance graphite dies and punches (electrically conductive) are used, a combination of alternating and direct current is applied, and a controlled vacuum atmosphere is required. A typical cycle of

SPS takes minutes of current passing to be completed. This does not result in cheap and attractive equipments and conditions for the industry. However, the ERS technique allows the use of durable electrically insulating dies, and the process can take place in the air, because a typical cycle is completed in seconds. Moreover, the necessary equipment can be easily adapted from the well-known resistance welding technology, therefore, being a much-tested technology.

Among the ERS advantages, in comparison with the conventional PM cold-pressing and furnace sintering route, the use of relatively low pressures (around 100 MPa) to achieve very high densities, the use of extraordinarily short processing times (around 1 s), and the possibility of operating in the air, without protective atmospheres, can be mentioned. Some recent works dealing with this modality can be found in [15–17]. In these works, a process conceptually similar to the ERS technique was successfully applied to iron-base materials, gold, silver tin oxide, titanium, or rare-earth magnets.

The medium-frequency electrical resistance sintering (MF-ERS) equipment used in this work brings an additional advantage: The medium-frequency technology allows the use of a direct current and the decrease of the weight and size of the welding transformer core, at the time that maintains its power. The main disadvantages of the ERS technique arise from operational difficulties (incomplete knowledge of how certain parameters influence the process) and the problem of achieving a homogeneous temperature distribution in the powder mass [18]. Some of the authors of this paper have studied this process, both from the experimental and theoretical points of view [19–21]. One of the important problems of the ERS technique refers to the durability of the ceramic dies, which was studied in [22].

In order to increase the knowledge about the ERS process, some aspects of the technique are analysed in this work. We will focus on the effect of secondary parameters as the applied pressure. For this purpose, a commercially pure iron powder has been electrically consolidated under various current intensity and heating time conditions, and two applied pressures (100 and 150 MPa). The study has been completed with the comparative analysis of the results obtained with 9 and 12 mm inner diameter dies, for equivalent processing conditions. Metallographic analyses of the samples processed by this route have been carried out, comparing the results with those obtained by the conventional PM route of cold pressing and furnace sintering. Literature has already studied a similar situation with this same material in [23].

## 2. Experimental Procedure and Materials

### 2.1. MF-ERS Equipment and Process

There is no commercial equipment specifically designed to perform the ERS experiments. However, the electrical requirements of the process (high intensity and low voltage) are adequately met by a resistance welding machine, which can also provide the mechanical load required for compression. To carry out the electrical consolidation experiments by MF-ERS, a properly adapted projection type resistance welding machine (Beta 214, Serra Soldadura S.A., Barcelona, Spain) was employed (Figure 1). This equipment, designed with the medium-frequency technology, incorporates a three-phase 1000 Hz and 100 kVA transformer, control electronics capable of providing a current output of a set value, and a servo-driven upper plate capable of providing a maximum load of 15 kN. The equipment is also conveniently instrumented to access the evolution of the relevant process parameters. Thus, the values of the current intensity, the voltage between the equipment plates and the position of the upper plate are monitored and recorded during the MF-ERS experiments.

In addition to the welding machine (the power and pressure source), the electrical consolidation process also requires a die containing the powders to be sintered, and the punches/electrodes to apply the pressure and electrical current (Figure 1). Following the design employed by Lenel [3], a die consisting of an alumina tube reinforced by a steel hoop, was used. Each punch/electrode consisted of a disk (wafer) of heavy metal tungsten alloy (75.3 wt% W–24.6 wt% Cu), with good nonstick and electro-erosion resistance properties, and a cylindrical bar made of temperature-resistant Cu-alloy

(98.9 wt% Cu, 1 wt% Cr, and 0.1 wt% Zr). The wafers also have lower thermal conductivity than the cylindrical bars, thus slowing down the heat leak from the compact to the water-cooled bedplates.

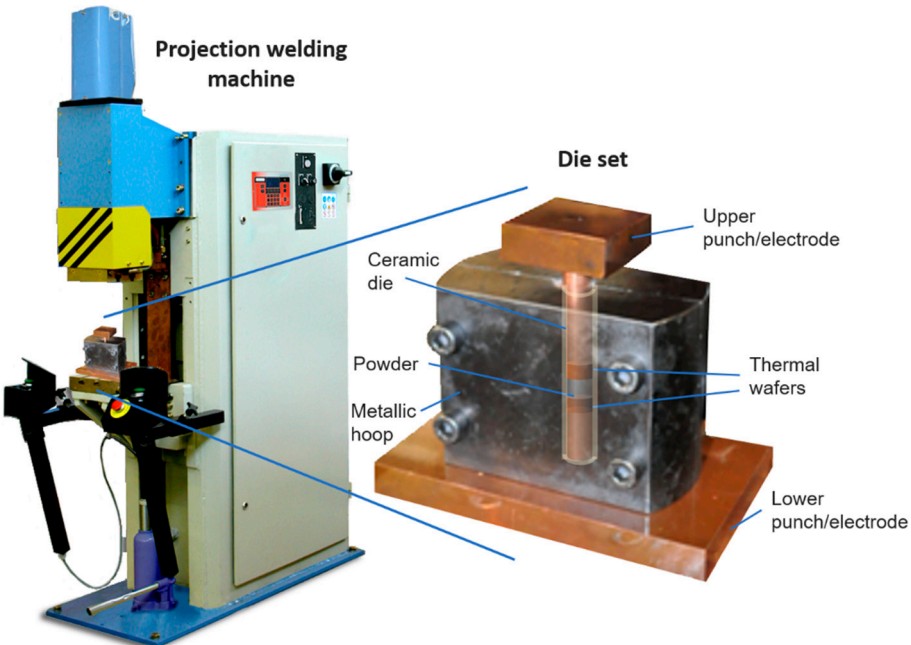

**Figure 1.** Electrical resistance welding machine adapted to act as the medium-frequency electrical resistance sintering (MF-ERS) equipment, and detail of the die set with a recreation of the interior.

Regarding the process, the inner wall of the ceramic tube was lubricated with graphite powder, deposited by means of a suspension of graphite in acetone. Once filled with powder, the die is shaken to make the powder to reach its tap porosity [24], thus ensuring the repeatability of the MF-ERS process.

MF-ERS experiments start with a cold pressing stage for 1000 ms. During this time, a constant pressure (100 or 150 MPa) is applied to the powder mass, but no current is passing through. The next step is a heating stage, with the only difference of current intensity being applied. The last step consists of a cooling and pressing stage for 300 ms, when again only pressure (100 or 150 MPa) is applied.

Electrical consolidation experiments were carried out with heating times of 400, 700, and 1000 ms and current intensities of 3.38, 4.50, and 5.63 kA (in case of using the 9 mm in diameter die). These intensities, normalised with the compact cross-section, represent current densities of 53.1, 70.7, and 88.4 A/mm$^2$. The same current densities are reached in experiments with the 12 mm die for current intensities of 6, 8, and 10 kA. The combination of lower intensities and/or heating times did not adequately consolidate the powder aggregate, whereas higher values welded the compact and wafers. In all the experiments, 3 g of powder were used, resulting in consolidated compacts with a mean aspect ratio (height/diameter) for the different processing conditions of approximately 0.85 for the 9 mm in diameter compacts, and 0.34 for those with 12 mm in diameter. Two different specimens were prepared for each of the tested conditions.

Moreover, compacts of 3 g, with 9 or 12 mm in diameter were conventionally consolidated: Firstly, cold compacted at 500 MPa (with lubricated die wall) and then, vacuum furnace sintered at 1175 °C for 30 min. Reducing atmospheres are required in order to reach low porosities, but selected conditions allow the comparison with the MF-ERS process carried out in air, without the chemical help of reducing atmospheres.

## 2.2. Material and Characterisation

The water-atomised starting powder, Fe WPL200 (QMP, Monchengladbach, Germany), contains 0.01 wt% C, 0.1 wt% O, and 0.2 wt% Mn as main impurities. This powder has an apparent density [25]

of 2.65 g/cm$^3$ (34% of the theoretical density), a tap density [24] of 2.79 g/cm$^3$ (35% of the theoretical one), and a mean particle size of D(4,3) = 88.2 m, determined by the laser diffraction technique (Mastersizer 2000, Malvern Panalytical Ltd., Malvern, UK). The SEM morphology (FEI Teneo, FEI Company, Hillsboro, OR, USA) of the powder is shown in Figure 2.

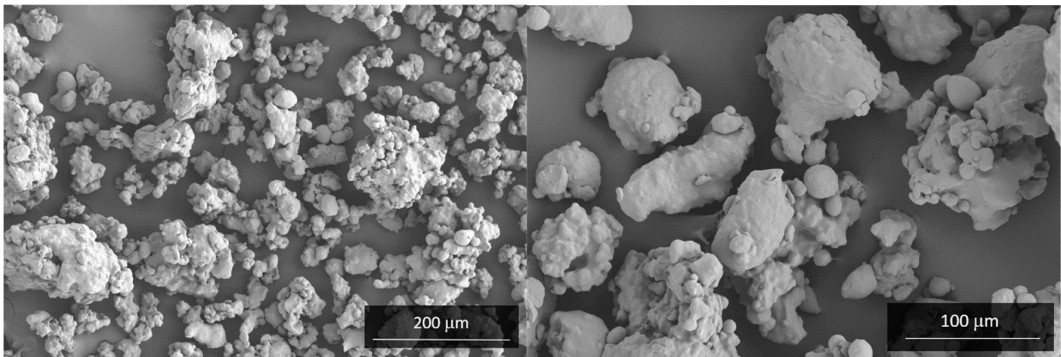

**Figure 2.** SEM micrographs of the Fe WPL200 powder used in this work.

As an additional characterisation of the powder, its compressibility curve [26] was experimentally determined (Figure 3).

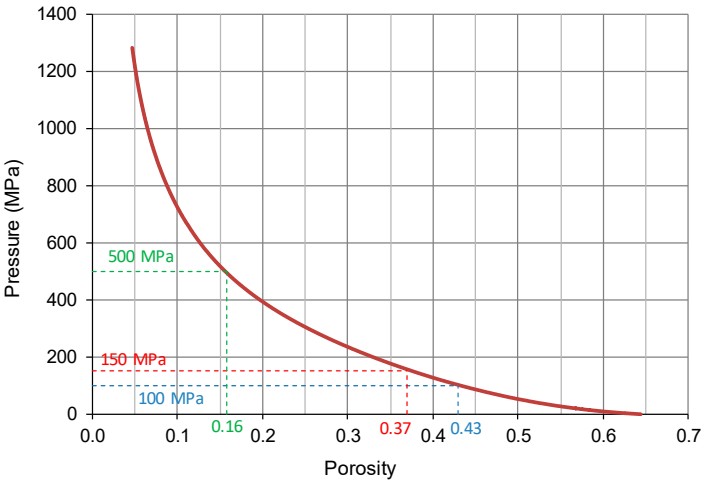

**Figure 3.** Compressibility curve of the Fe WPL200 powder.

From this curve, and for the chosen applied pressures (100 and 150 MPa), the initial porosities are 0.43 and 0.37, respectively. The pressure applied for the conventional PM route (500 MPa) leads to a green porosity of 0.16.

Diametrical sections of the compacts (electrically and conventionally consolidated) were macroscopically analysed to know their porosity distribution. The found nonhomogeneous porosity distribution is a consequence of the temperature reached at different zones during the MF-ERS process. On the other hand, the global porosity after the MF-ERS experiments (i.e., in the final situation after finishing the consolidation process) was calculated from the dimensions (diameter and height measured at eight different positions for each specimen) and weight of two different specimens for each experimental condition (according to the uncertainties in the measurements, the porosity values determined by this method may have a 5% uncertainty).

The vickers microhardness (applied load of 1 kg according to [27]) on MF-ERS compacts was measured in a diametrical-section quadrant in five different points, as shown in Figure 4 (different positions are tested because of the nonuniform porosity distribution, all the measured values were

averaged to determine the mean microhardness). Other quadrants are supposed to behave in a similar way due to the symmetry of the compact.

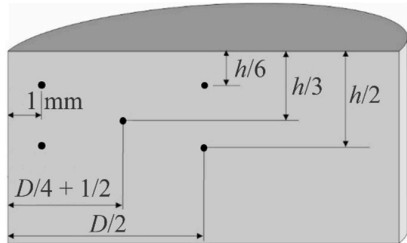

**Figure 4.** Microhardness indentation map on a MF-ERS compact diametrical section.

## 3. Results and Discussion

### 3.1. Effect of Processing Parameters on Porosity

Porosities (Θ) of the electrically consolidated compacts are shown in Table 1, as a function of the applied pressure (100 or 150 MPa), current intensity (*I*), and heating time ($t_H$).

Porosities in Table 1 follow the expected trend, decreasing downwards and to the right of the Table for both applied pressures. For a better analysis, data in Table 1 have been represented in Figure 5.

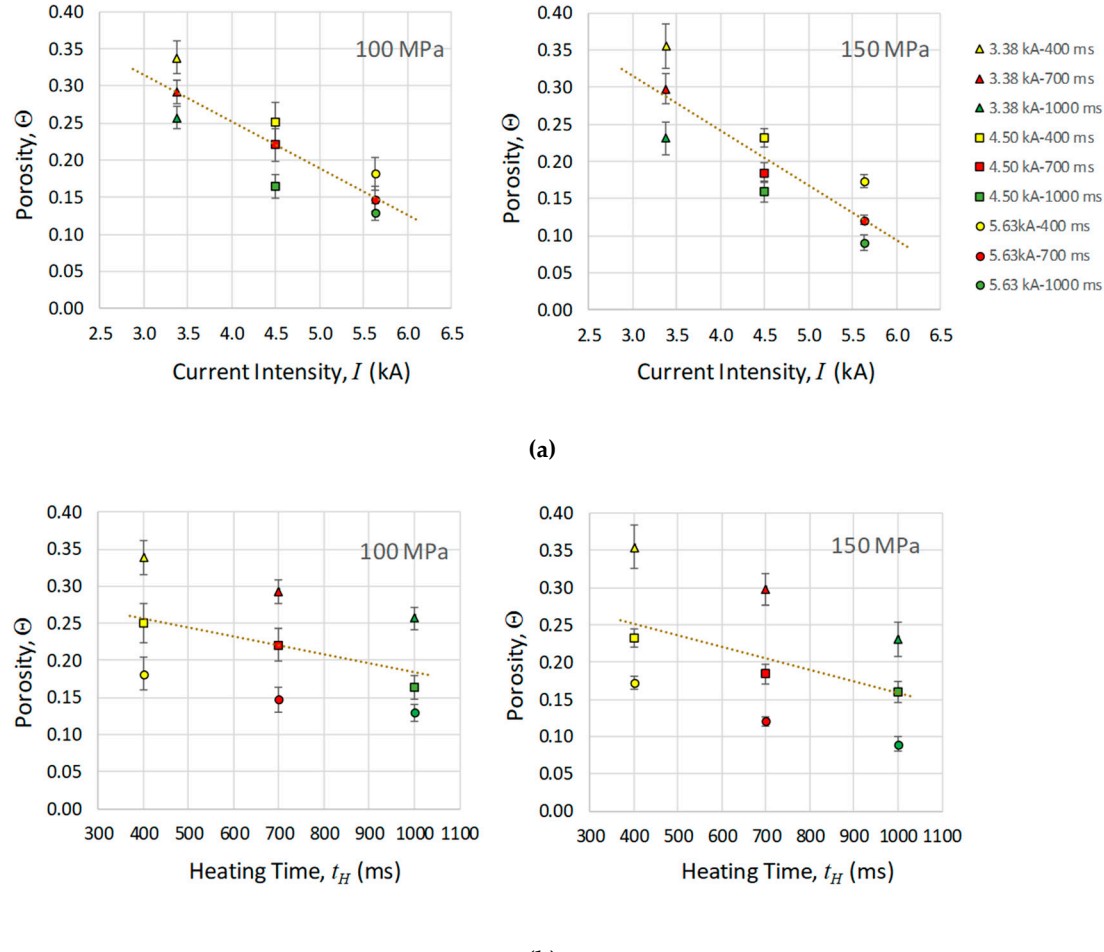

**(a)**

**(b)**

**Figure 5.** Porosity of 9 mm in diameter compacts (Θ) represented versus: (**a**) Current intensity and (**b**) heating time, for the different MF-ERS experiments with two pressures (100 and 150 MPa). The dotted lines in the graphs represent the trend of the whole set of points.

**Table 1.** Porosities (Θ) for different electrical consolidation conditions for 9 mm in diameter compacts.

| Processing Conditions | | Porosity, Θ | |
| --- | --- | --- | --- |
| | | Pressure (MPa) | |
| Intensity (kA) | Heating Time (ms) | 100 | 150 |
| | 400 | 0.34 | 0.36 |
| 3.38 | 700 | 0.29 | 0.30 |
| | 1000 | 0.26 | 0.23 |
| | 400 | 0.25 | 0.23 |
| 4.50 | 700 | 0.22 | 0.18 |
| | 1000 | 0.16 | 0.16 |
| | 400 | 0.18 | 0.17 |
| 5.63 | 700 | 0.15 | 0.12 |
| | 1000 | 0.13 | 0.09 |

Figure 5 shows that, in general, for any couple of intensity/time values, the porosity is slightly lower when the applied pressure is higher. However, the effect is not very significant and must be relativised. In fact, the process is more efficient for the lower pressure (higher initial porosity), with an average porosity reduction of 49%. A 45% of porosity reduction is reached for 150 MPa. Moreover, clear trends parallel to the global trend are shown, both for current intensity and heating time families. The comparison of the graphs indicates that, for the studied values, current intensity has a greater influence on the compacts porosity than does the heating time.

### 3.2. Relationship between Specific Thermal Energy and Porosity

The Joule thermal energy released per powder unit mass, which will be called the specific thermal energy (*STE*), can be computed by integrating the dissipated electric power during the heating time (*t*). That is,

$$STE = \frac{1}{M} \int_{0}^{t_H} V(t) \cdot I(t) dt \qquad (1)$$

where *M* is the powder mass, *I* is the current intensity that flows through the powder aggregate, and *V* is the voltage drop in the powder column. The experimental uncertainty for the *STE* determination has been estimated in 3%. However, as with the porosity, the *STE* values shown in Table 2 are the result of the arithmetic mean of the values obtained in the two experiments carried out under identical conditions. Considering the deviations from the mean value, the uncertainty increases to 4%.

*STE* follows the expected behaviour, with greater values for higher intensities and heating times. According to Table 2, STE values for 150 MPa are lower than those corresponding to 100 MPa of pressure. This is because the initial porosity for 150 MPa is lower, and consequently, the resistivity of the powder mass is also lower. Since the power released by the Joule effect is linearly proportional to the resistivity, the *STE* will be lower for higher applied pressures. Thus, lower porosities are achieved for 150 MPa just because the initial porosity is lower, despite the thermal energy released in the compact (and the maximum temperature reached) is lower.

Figure 6 shows the *STE* vs. porosity (Θ) scatter plot, corresponding to the two applied pressures (also shown are the linear trends). It can be seen, as previously commented, that the *STE* is higher for lower porosities. Moreover, it is clear how the released *STE* is higher when the applied pressure is lower.

**Table 2.** Specific thermal energy (*STE*) values (expressed in kJ/g) for the different MF-ERS experiments.

| Processing Conditions | | Specific Thermal Energy, *STE* | |
| :---: | :---: | :---: | :---: |
| | | Pressure (MPa) | |
| Intensity (kA) | Heating Time (ms) | 100 | 150 |
| | 400 | 0.37 | 0.32 |
| 3.38 | 700 | 0.52 | 0.43 |
| | 1000 | 0.58 | 0.56 |
| | 400 | 0.56 | 0.55 |
| 4.50 | 700 | 0.64 | 0.59 |
| | 1000 | 0.80 | 0.65 |
| | 400 | 0.70 | 0.60 |
| 5.63 | 700 | 0.78 | 0.73 |
| | 1000 | 0.82 | 0.83 |

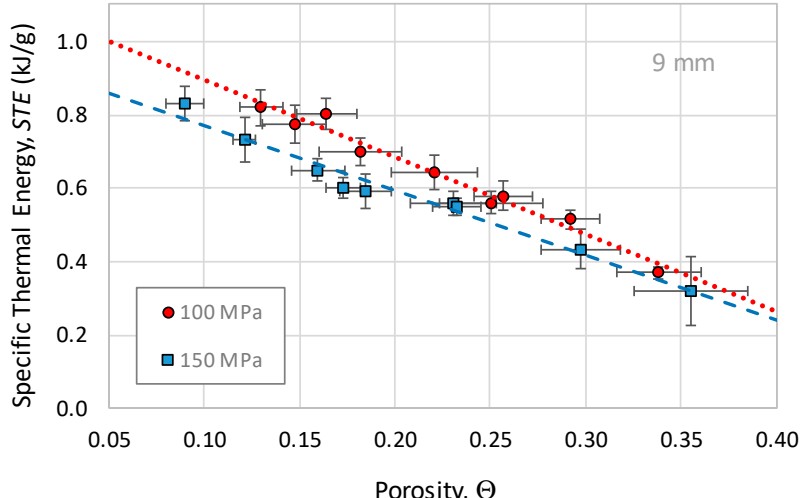

**Figure 6.** Specific thermal energy versus porosity (Θ) for both applied pressures and different intensities and current passing times in 9 mm in diameter compacts. Lines represent the trend of the two sets of points.

Results obtained with the 9 mm die can now be compared with those for the 12 mm die for 100 MPa of applied pressure. A total of 3 g of powder and identical current densities (53.1, 70.7, and 88.4 A/mm$^2$) were used in both cases, although compacts with different aspect ratios were obtained. Figure 7 shows the *STE* vs. Θ scatter plot.

The first conclusion to be drawn is that the trend line for 9 mm compacts is on the right side or above the one for 12 mm. Therefore, to achieve a certain Θ with the 9 mm die, a higher *STE* is required. In other words, for a *STE*, lower densification is reached with the 9 mm die. Regarding this conclusion, it has firstly to be noted that the sum of the lateral and basal areas for 12 mm compacts is approximately 13% greater than that for 9 mm (considering mean values of the initial and final height of the compacts, i.e., aspects ratios of 0.45 and 1, for 12 and 9 mm, respectively). Thus, heat leaks should be apparently greater for 12 mm compacts. In second place, 12 mm compacts should feel a thermal gradient in vertical direction twice that of 9 mm compacts, because of their lower aspect ratio. Thermal leaks should, therefore, be again higher for 12 mm compacts. (Note that the vertical direction is considered because the wafers and electrodes are more conductive than the die, and are in contact with cooled plates.) It can finally be stated that larger leaks mean reaching lower temperatures, and

consequently higher porosity. According to this, points corresponding to 12 mm compacts in Figure 7 should be for a *STE* on the right of those for 9 mm compacts, i.e., the opposite to that observed. Thus, if lower porosities are attained with the 12 mm die, with higher heat leaks, a new factor should be taken into account. The reason can be probably found in the average pressure within the compacts ($\overline{P}$), which, for single action press processes as taking place in MF-ERS, decreases when the compact aspect ratio increases, according to [28]:

$$\overline{P}/P = 1 - 2\mu z(H/D) \tag{2}$$

where $P$ is the applied pressure, $\mu$ is the kinetic friction coefficient, $z$ is the radial-to-axial pressure ratio, and $H/D$ is the aspect ratio (height/die diameter).

Considering a friction coefficient between powders and die walls of 0.25 [29], and a radial-to-axial pressure ratio of 1 (pseudo-hydrostatic conditions), the average internal pressure decreasing factor (Equation (2)) reaches up to about 0.78 for 12 mm compacts, and 0.5 for 9 mm. Therefore, the average internal pressure can decrease more in high aspect ratio 9 mm compacts for the same applied pressure, increasing the porosity. Considering the short duration of the MF-ERS process and the high contact thermal resistances, the thermal leaks influence will be low, and the obtained results seem to be expected.

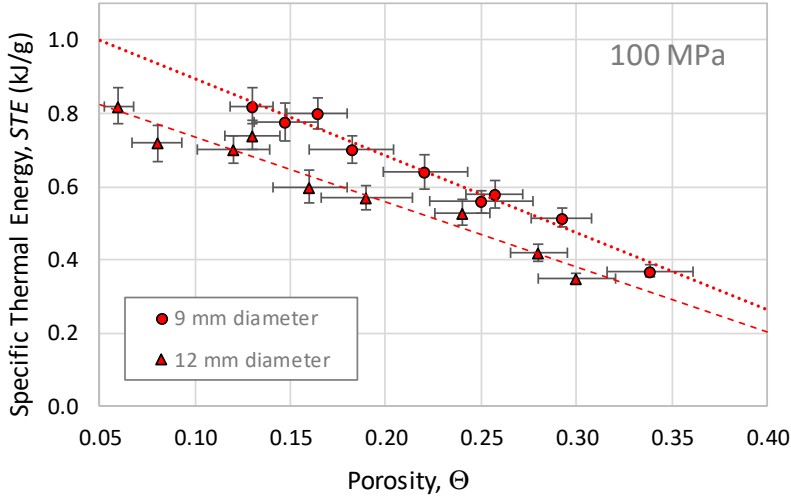

**Figure 7.** Specific thermal energy versus porosity (Θ) in experiments carried out with 100 MPa of applied pressure and the same current densities and powder mass, but different inner diameters of the die (9 and 12 mm). Lines represent the trend of the two sets of points.

## 3.3. Porosity Distribution inside the Compacts

The nature of the MF-ERS process, mainly because of its high speed, produces micro- and macro-structural characteristics very different to those observed in the compacts obtained by other FAST techniques and the conventional PM route. Figure 8 shows diametrical sections of conventionally sintered and of electrically consolidated compacts under the different combinations of current intensity, heating time, and applied pressure. The conventional compact shows a relatively uniform porosity of about 15%. In contrast, the MF-ERS compacts reveal a less uniform porosity distribution, due to the existence of temperature gradients during the consolidation process. Thus, the compact core reaches a higher temperature than the peripheral areas (cooled by the heat leaks towards the die walls and the cooled electrodes), therefore, achieving a higher densification.

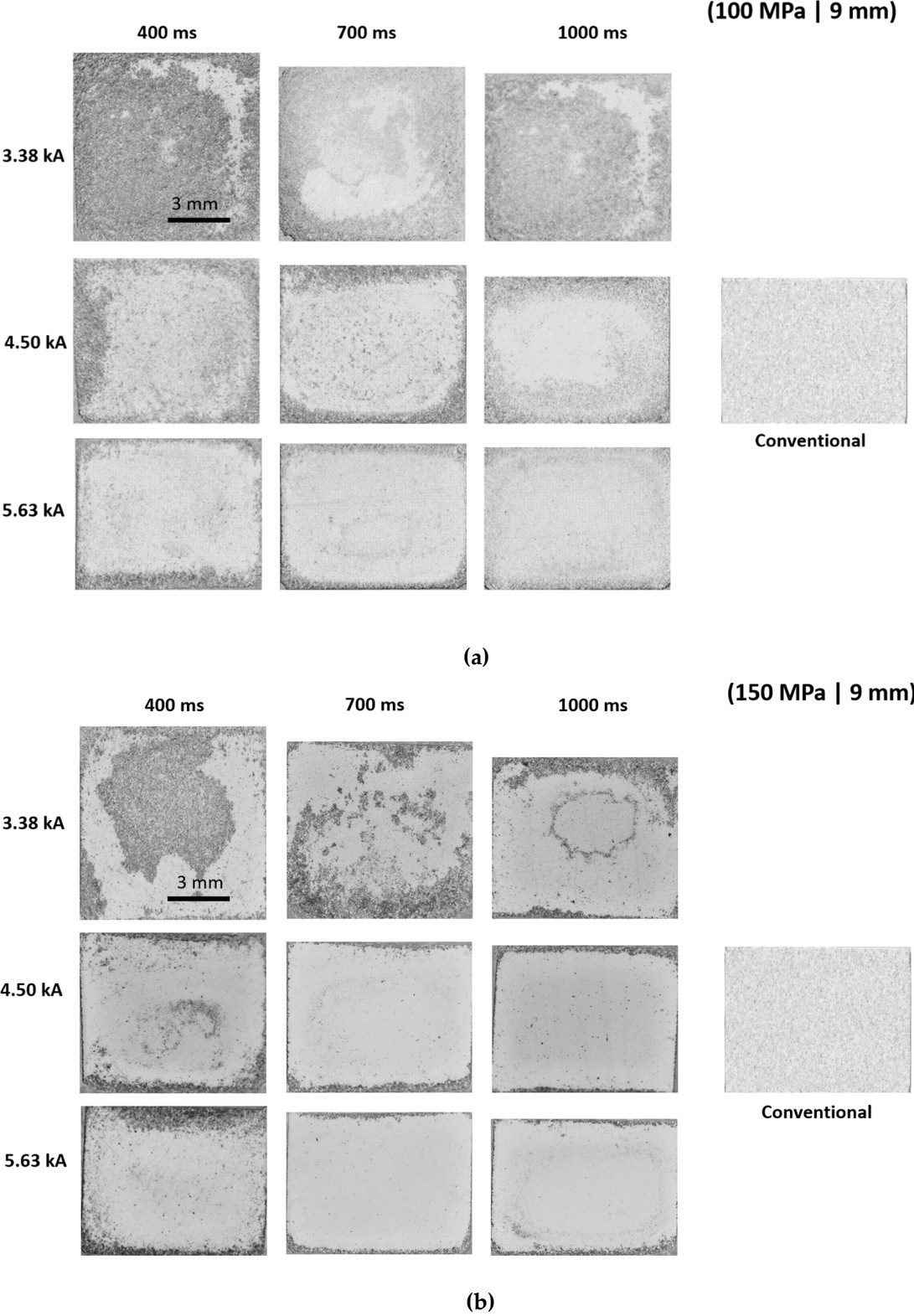

**Figure 8.** Porosity distribution in 9 mm MF-ERS compacts for an applied pressure of (**a**) 100 and (**b**) 150 MPa. The macrograph of the compact obtained by the conventional PM route is added for comparison. In all the images, clearer areas (more reflecting zones) indicate a lower porosity. Current flows in all cases from the top to the bottom of the images.

Comparing the macrographs sequence obtained for the two applied pressures, the porosity is higher when the applied pressure is lower. For this pressure (100 MPa), a uniform distribution of porosity is practically only attained for the last condition, with high intensity and current passing time. On the other hand, with 150 MPa, the starting situation is already more uniform, and for 3.38 kA/700 ms, a quite uniform distribution of porosity is achieved. Nevertheless, thin layers of porous material are still obtained, mainly in the contact with the cooled wafers/electrodes.

It is worth considering whether it is possible or not to get higher densifications with the MF-ERS technique. It must be considered that the studied range of current intensities and heating times was limited by the weld of the compact to the wafers under more severe conditions. Nevertheless, for these conditions, the inner of the compact did not melt (something that can take place for very severe conditions [20]). Therefore, for any of the studied pressures, the current intensity or heating time can be increased if the nonstick properties of the wafers are improved. This could be achieved with an improved alloy or, more easily, with a thin layer of graphite on the wafer surface in contact with the powder.

### 3.4. Microhardness

The hardness of porous PM materials depends mainly on two factors: Inversely with the degree of porosity, and directly and with a weaker influence with the goodness or quality of the bond between particles, i.e., the sintering quality [30,31]. Regarding the first factor, and given the nonuniform distribution of porosity in MF-ERS compacts, higher microhardness values can be expected in the centre of the compact (the densest zone) and lower values in the periphery (the most porous zone). In order to simplify the following analysis, only the mean values and deviations from the five measurements in the two specimens prepared for each processing condition will be considered. The mean microhardness values and their deviations are shown in Table 3.

**Table 3.** Microhardness (HV1) mean values and standard deviations calculated from five measurements at the different 9 mm MF-ERS compacts.

| Processing Conditions | | Microhardness, HV1 | |
|:---:|:---:|:---:|:---:|
| | | Pressure (MPa) | |
| Intensity (kA) | Heating Time (ms) | 100 | 150 |
| | 400 | 30 ± 4 | 24 ± 8 |
| 3.38 | 700 | 44 ± 13 | 36 ± 7 |
| | 1000 | 46 ± 18 | 55 ± 16 |
| | 400 | 46 ± 3 | 54 ± 12 |
| 4.50 | 700 | 56 ± 9 | 65 ± 21 |
| | 1000 | 70 ± 8 | 73 ± 20 |
| | 400 | 65 ± 13 | 70 ± 16 |
| 5.63 | 700 | 75 ± 16 | 85 ± 12 |
| | 1000 | 87 ± 10 | 95 ± 19 |

As expected, the mean microhardness increases with the current intensity and heating time. On the other hand, the comparison for the two applied pressures shows that microhardnesses are generally higher for the applied pressure of 150 MPa. This can be caused by the slightly lower porosity reached for the higher pressure. Nevertheless, as aforementioned, the main factor affecting microhardness is porosity, and a direct relationship between both properties is expected, as shown in Figure 9. For an intensity and time in Table 3, the couple of values shown are not obtained for the same porosity, because a different pressure is applied.

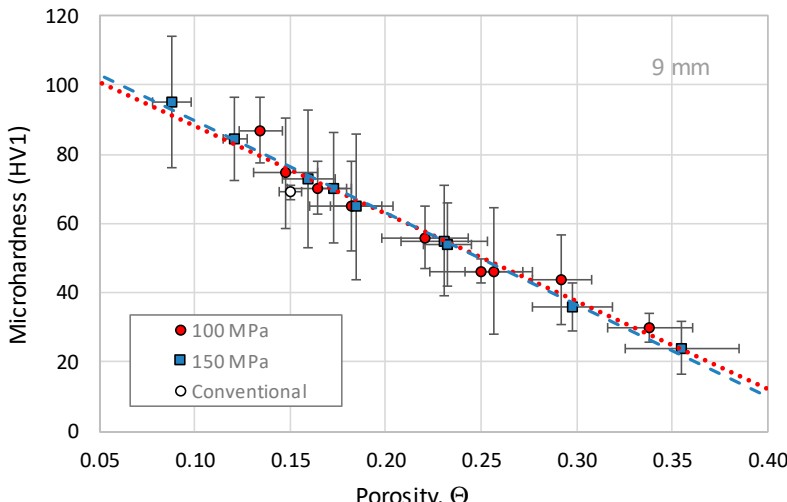

**Figure 9.** Mean microhardness versus porosity ($\Theta$) for 9 mm compacts and the two applied pressures. Lines represent the trend of the two sets of points. The microhardness value of the conventionally consolidated compact is included for comparison.

In addition, deviations are higher for 150 MPa. For 100 MPa, deviations represent between 7% and 39% of the microhardness value, with an average value of 19%. For 150 MPa, these figures are between 17% and 40%, with an average value of 28%. The reason for this higher deviation in the 150 MPa processed compacts, with lower porosity, must be the bond quality between particles, i.e., the sintering quality, which must be worse for these compacts. This is probably due to the lower *STE* released for higher pressures (see Table 2).

The mean microhardness of the conventionally processed specimen resulted in 69 ± 5 HV1, an expected value according to its porosity (its icon in Figure 9 appears very near to the data points of the MF-ERS compacts). Nevertheless, for the tested conditions, the electrical process allows reaching higher densities, and this results in higher values of microhardness. In order to reach higher densities and microhardnesses with the conventional PM method, reducing atmospheres and/or higher sintering temperatures should be tested. However, this could decrease the hardness by increasing the grain size or by relaxing the manufacturing (sudden cooling) internal stresses of the starting powder. Obviously, the electrical consolidation process shows a clear advantage due to the very short exposition to high temperatures, only slightly altering the initial microstructure of the powders.

Finally, Figure 10 shows the microhardness mean values measured in 9 and 12 mm compacts. Considering the data dispersion, results shows similar trends, with the line for 12 mm compacts slightly under that for 9 mm compacts. This agrees with results in Figure 7; for any particular value of porosity, the released *STE* is higher for the 9 mm compacts, leading to a better bond between particles, and contributing to a slightly higher microhardness. Nevertheless, differences are attenuated for low porosities, therefore, being the effect of the bonding quality on the mean microhardness lower for denser materials (although this is not the behaviour for the deviations values).

It is also worth noting that the several parameters studied in this work, and their effect on the MF-ERS compacts properties, are difficult to follow when more than one parameter affect the results. The only way to validate the certain hypothesis used to explain the results is the support of numerical simulations. Thus, it would be desirable to predict the temperature at the contact compact/wafer, being then possible to increase the severity of the working conditions and the densification, at the time that avoiding the weld between compact and wafers. This will be the work to be carried out, specifically for the conditions of this manuscript, in the near future.

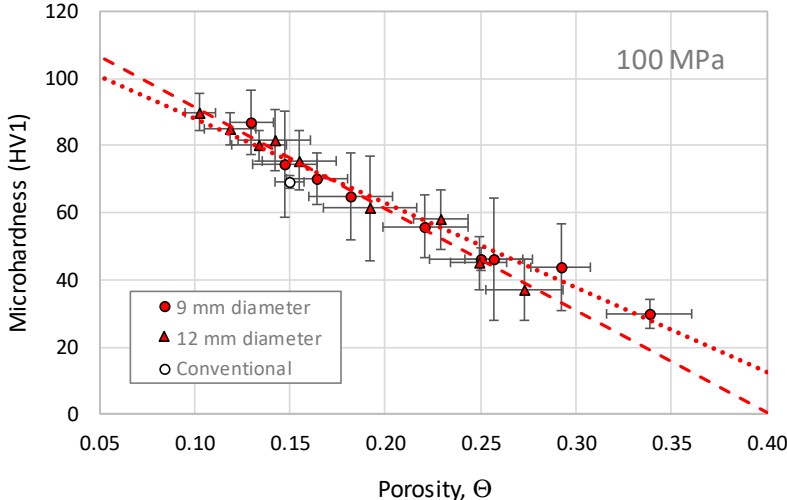

**Figure 10.** Mean microhardness versus porosity ($\Theta$) in experiments carried out with 100 MPa of applied pressure and the same current densities and powder mass, but different inner diameters of the die (9 and 12 mm). Lines represent the trend of the two sets of points.

## 4. Conclusions

In this work, iron powder has been successfully consolidated by means of the MF-ERS process, within the window of analysed conditions (varying the current intensity, heating time, and applied pressure). The densification achieved with this technique, as well as the attained microhardness values, are equal or higher than those obtained by the conventional PM route, with the added advantage of an extraordinarily short duration (around 1 s as opposed to 30 min) and the use of lower pressures (100 or 150 MPa as opposed to 500 MPa).

The lowest and most uniform porosity, and the highest microhardness, is achieved for both tested pressures with the combination 5.63 kA/1000 ms. Lower porosities are achieved with 150 MPa than with 100 MPa (although the effect is lower than expected). The expected correlation between mean microhardness and porosity (i.e., higher microhardness for lower porosities) is also clear. However, the higher applied pressure does not result in greater uniformity of microhardness measurements. This can be explained by the lower thermal energy released when the applied pressure is higher, producing a poorer bond between particles that affect measurements deviations. Finally, it can be concluded that a lower aspect ratio in the compact allows achieving denser materials.

Future work regarding this study includes the numerical simulation of the results shown in this manuscript. The obtained prediction will be useful to improve the knowledge of the technique, to check the hypothesis now considered, and to improve the technique by reaching lower porosities.

**Author Contributions:** Conceptualization, J.M.M. and F.G.C.; methodology, J.M.M. and F.G.C.; validation, R.A., F.T., and E.S.C.; writing—original draft, F.T.; writing—review and editing, F.G.C., J.M.M., and F.T. All authors have read and agreed to the published version of the manuscript.

**Funding:** This research was funded by Ministerio de Economía y Competitividad (Spain) and Feder (EU) through the research projects DPI2015-69550-C2-1-P and DPI2015-69550-C2-2-P.

**Acknowledgments:** The authors also wish to thank the technicians J. Pinto, M. Madrid, and M. Sánchez (University of Seville, Spain) for experimental assistance.

**Conflicts of Interest:** The authors declare no conflict of interest.

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
