# Peer review of "Influence of Processing Parameters on the Conduct of Electrical Resistance Sintering of Iron Powders"

_metals, doi:10.3390/met10040540_

Round 1
Reviewer 1 Report
There are some minor errors in the article, for example:
- line 14: “…consolidated by the Medium…” instead ““…consolidated be the Medium…”,
- line 50: “…use on the industrial…” instead “…use on an industrial…”,
- lines 56 and 62: “…in the air…” instead “…in air…”,
- line 65: “…time that is maintaining…” instead “…time that maintaining…” and so on.
I recommend a lingual review of the entire document.
The title of the article isn’t clear; my suggestion is as follows: Influence of Processing Parameters on the Conduct of the Electrical Resistance Sintering of Iron Powders
There is the statement in the abstract: “The influence of the applied pressure and electrical parameters on the microstructure of specimens…”. Where are the pictures of microstructure? Figure 8 presents only porosity distribution in the samples. You should place the appropriate microstructure photos and describe them.
There is the sentence in the 103-104 lines: “Once filled… ….of the results.” This is not clear for me. Could you explain them?
There are presented (lines 119-121) parameters of conventional process. Authors don’t justify why the indicated conditions were used, for example: 500 MPa for compacting, type of sintering atmosphere, temperature, times end so on. The authors should explain if these conditions are optimal for this producing process. Maybe they are the result of previous experiments?
The photos in the Figure 8 are too small and not clear. You should insert bigger ones or possibly insert only few of them and describe significant differences.
Author Response
Dear reviewer, thank you for detailed review and your interest in improving the content of this manuscript. Please find next our answer to your different comments. We hope the changes included in the manuscript are in accordance with your suggestions.
There are some minor errors in the article, for example:
line 14: “…consolidated by the Medium…” instead ““…consolidated be the Medium…”,
line 50: “…use on the industrial…” instead “…use on an industrial…”,
lines 56 and 62: “…in the air…” instead “…in air…”,
line 65: “…time that is maintaining…” instead “…time that maintaining…” and so on.
I recommend a lingual review of the entire document.
A proof-read has been performed in the manuscript, and changes have been included in the new version. The sentences you comment have been corrected.
The title of the article isn’t clear; my suggestion is as follows: Influence of Processing Parameters on the Conduct of the Electrical Resistance Sintering of Iron Powders
The title has been changed according to your suggestion.
There is the statement in the abstract: “The influence of the applied pressure and electrical parameters on the microstructure of specimens…”. Where are the pictures of microstructure? Figure 8 presents only porosity distribution in the samples. You should place the appropriate microstructure photos and describe them.
Also according to other reviewer suggestion, any reference to microstructure has been avoided in this new version of the manuscript. We are indeed studying the macrostructure of these sintered compacts, and that is now indicated in the text.
There is the sentence in the 103-104 lines: “Once filled… ….of the results.” This is not clear for me. Could you explain them?
We have changed the end of the sentence, instead of “results” now it can be read “MF-ERS process”. We meant that powders are not just poured inside the die, because this is not a standardised process, but the die is filled and the powders inside prepared to the MF-ERS process according to MPIF Standards. This ensures the repeatability of the process, and as a consequence, of the results, as we had indicated in the previous version.
There are presented (lines 119-121) parameters of conventional process. Authors don’t justify why the indicated conditions were used, for example: 500 MPa for compacting, type of sintering atmosphere, temperature, times end so on. The authors should explain if these conditions are optimal for this producing process. Maybe they are the result of previous experiments?
Regarding the conventional PM process of cold-pressing + furnace-sintering, conditions were not selected to optimize the process (reducing atmospheres are a need for such purpose, or even double cycles of cold pressing + furnace sintering). The conditions were selected according to usual values in Fe sintering processes, and mainly, to show the densification capacity of the conventional process without chemical help. This makes fair the comparison with the electrical route, which does not use this chemical help. We have included a sentence to clarify this in the text: “Reducing atmospheres are required in order to reach low porosities, but selected conditions allow the comparison with the MF-ERS process carried out in air, without the chemical help of reducing atmospheres.”
The photos in the Figure 8 are too small and not clear. You should insert bigger ones or possibly insert only few of them and describe significant differences.
We consider that it is necessary to show all of the photos in order to clearly see the general behaviour of the process with respect to the processing parameters. On the other hand, the intention is not to show microscopic details, but, mainly, the different areas in the compacts and a general idea of the consolidation reached in the centre and the compact periphery.
Nevertheless, we are open to change and increase the size of the photos, but we would like that all of them were finally in the same page to make easier the comparison. Thus, if finally the manuscript is accepted, and the editorial office finds during the final preparations for publication the way to rearrange the Figure with larger images, it will be fine for us to change it.

Reviewer 2 Report
This work is a systematic study of various parameters used in MF-ERS of iron powder. There have not been many other studies done on MF-ERS, so this study is important to the metal sintering community. However, there are several issues with the writing and figures which needs to be corrected before further consideration.
- Several major English mistakes were found. Please perform a thorough proof-read. Here are just a few which I found:
- The title should be “Effect of processing parameters on the….” Not processing parameters effect
- Line 21 – “Experiences” is an incorrect word choice.
- Line 24 – it should be higher “current” intensities
- Line 119 – “compacts of 3 g, and 9 and 12 mm in diameter”. This sentence needs clarification.
- Lines 332-333 – I am very confused by this sentence as it does not make much sense. I think the sentence needs to be rephrased.
- Introduction
- The first three paragraphs of the introduction are about the historical background of sintering metallic powders. I think it can be concisely summarized into a single paragraph since it is not very important.
- The introduction could also include some information on the previously studied MF-ERS systems
- How were the current intensity values determined? It seems like such random values were selected. For example, 3.38, 4.5 and 5.63 kA.
- All the tables do not have the proper heading. For example, Table 1 does not state that the column values are pertaining the final porosities. The reader should not have to read the caption to figure out what is contained inside the table. Please correct this issue for all the tables. I understand that there are many parameters involved in this study, but it seemed weird to separate the values by a slash. I would suggest for a better way to present your data in the tables.
- Please make sure to include error bars for the measurements. All of figures did not have any error bars. It is impossible to compare the results between samples and conditions without proper statistical analysis.
- Figure 8 should have markers on where the electrode locations are (is it on the top/bottom or left/right?)
- Some suggestions on how to avoid the cooling effects of electrodes would be greatly appreciated in the manuscript. This effect also seems to be much more evident in lower current intensities, but not in the higher current intensities. Why is that?
- For the microhardness testing, the authors commented on the distribution of microhardness due to variation in porosity in different areas. How did the authors then perform the 5 indentations between two samples? Where are they located and will that be a fair comparison between the samples?
- In section 3.4 and also the conclusion, several statements discussing “bonding” as a reason for difference in microhardness. I believe the authors are referring to “grain/particle connectivity” rather than bonding. Bonding cannot be the reason for lower/higher hardness since a pure metal will only have metallic bonding. The particle connectivity is directly correlated to the porosity and the grain size. There was not any analysis of the grain size, so the data can only be supported by porosity data.
Author Response
Dear reviewer, thank you for detailed review and your interest in improving the content of this manuscript. Please find next our answer to your different comments. We hope the changes included in the manuscript are in accordance with your suggestions.
- Several major English mistakes were found. Please perform a thorough proof-read. Here are just a few which I found:
- The title should be “Effect of processing parameters on the….” Not processing parameters effect.
- Line 21 – “Experiences” is an incorrect word choice.
- Line 24 – it should be higher “current” intensities
- Line 119 – “compacts of 3 g, and 9 and 12 mm in diameter”. This sentence needs clarification.
- Lines 332-333 – I am very confused by this sentence as it does not make much sense. I think the sentence needs to be rephrased.
A proof-read has been performed in the manuscript, and changes have been included in the new version. The sentences you comment have been corrected.
- Introduction
- The first three paragraphs of the introduction are about the historical background of sintering metallic powders. I think it can be concisely summarized into a single paragraph since it is not very important.
- The introduction could also include some information on the previously studied MF-ERS systems.
Regarding the first point, we have a problem, because one of the reviewers asks to make the opposite (“The introduction is quite exhaustive and well-explains the context of the work. However, some historical description should be better described, focusing on the process characteristics and main improvements of ERS during time”). The only option we find is to maintain the same content as was initially. Sorry about this.
We have however added information of other studies carried out on MF-SRE by other authors. In particular the following new references have been added:
- Fais, A., A Faster FAST: Electro-Sinter-Forging. Metal Powder Report. 2018, 73(2), 80-86.
- Cannella, E.; Nielsen, C.V.; Bay, N. Process Investigation and Mechanical Properties of Electro Sinter Forged (ESF) Titanium Discs. The International Journal of Advanced Manufacturing Technology. 2019, 104, 1985–1998.
- Brisson, E; Carre, P; Desplats, H; Rogeon, P; Keryvin, V; Bonhomme, A. Effective Thermal and Electrical Conductivities of AgSnO2 During Sintering. Part I: Experimental Characterization and Mechanisms. Metallurgical and Materials Transactions A. 2016, 47, 6304–6318.
- Fais, A; Actis Grande, M; Forno, I. Influence of Processing Parameters on the Mechanical Properties of Electro-Sinter-Forged Iron Based Powders. Materials & Design 2016, 93(5), 458-466.
- How were the current intensity values determined? It seems like such random values were selected. For example, 3.38, 4.5 and 5.63 kA.
Please, note that this is explained in the original version of the manuscript at the end of section 2.1. We initially used 6, 8 and 10 kA for the 12 mm die. Then, in order to maintain the same current densities, the current intensities for the 9 mm die were determined, resulting in 3.38, 4.5 and 5.63 kA.
- All the tables do not have the proper heading. For example, Table 1 does not state that the column values are pertaining the final porosities. The reader should not have to read the caption to figure out what is contained inside the table. Please correct this issue for all the tables. I understand that there are many parameters involved in this study, but it seemed weird to separate the values by a slash. I would suggest for a better way to present your data in the tables.
We have tried to correct this issue in the new tables. Each value contained in the table is now in a separate cell, and table headings have been added.
- Please make sure to include error bars for the measurements. All of figures did not have any error bars. It is impossible to compare the results between samples and conditions without proper statistical analysis.
We had indicated the error for measurements in the text in a global way (for instance, Section 2.2 indicated the maximum error in the determination of the porosity, being a 5%). According to the reviewer suggestion, we have now include the error bars in the graphs.
- Figure 8 should have markers on where the electrode locations are (is it on the top/bottom or left/right?)
We thought it was clear because the different height of the images are caused by the press and current passing from the top to the bottom. We do not see clear how to include this in the figure, therefore we have preferred to indicate this in the figure caption in this new version.
- Some suggestions on how to avoid the cooling effects of electrodes would be greatly appreciated in the manuscript. This effect also seems to be much more evident in lower current intensities, but not in the higher current intensities. Why is that?
Unfortunately, there are no suggestions on how to avoid the cooling effect, because heat is generated inside the compact and it flows to cooler zones. We are already using wafers to avoid compacts getting stuck to the electrodes, but even with this conditions the electrodes have to be refrigerated, and heat leaks and their effect on the porosity is a characteristic of this technique.
Regarding its relationship with the current intensity, we consider that the effect is not more evident for low intensities. We guess you indicate this after viewing the macrographs in Fig 8, however, the observed structure for low intensities is caused by the lower thermal energy released inside the compact, not because of a higher cooling effect.
- For the microhardness testing, the authors commented on the distribution of microhardness due to variation in porosity in different areas. How did the authors then perform the 5 indentations between two samples? Where are they located and will that be a fair comparison between the samples?
The five indentations were not performed between two samples. We repeated the process (five measurements) in the two different samples processed under the same conditions. Therefore, the microhardness for any particular condition of pressure, current intensity and heating time is the mean value of 10 measurements. The location of this measurements in each one of the specimens is indicated in Figure 4.
- In section 3.4 and also the conclusion, several statements discussing “bonding” as a reason for difference in microhardness. I believe the authors are referring to “grain/particle connectivity” rather than bonding. Bonding cannot be the reason for lower/higher hardness since a pure metal will only have metallic bonding. The particle connectivity is directly correlated to the porosity and the grain size. There was not any analysis of the grain size, so the data can only be supported by porosity data
We were referring all these times to the bonding caused by the sintering process, and we agree that this can be confusing and related to the atomic bond. However, we consider the term bonding is correct. Please, note for instance that the ASM Handbook Vol 7: Powder Metallurgy, in Section 2 Terms and Definitions, says:
Bonding. The joining of compacted or loose particles into a continuous mass, under the influence of heat.
We therefore consider that the term is correct, and the concept is already explained in the second and third lines of section 3.4. If the reviewer still considers that this has to be clarified we are open to include some more explanations.

Reviewer 3 Report
This manuscript deals with an interesting topic. The conclusions at the end of the manuscript are sound and well presented.
The present structure of the manuscript is:
Abstract
1. Introduction
2. Experimental Procedure and Materials
2.1. MF-ERS equipment and process
2.2. Material and characterisation
3. Results and Discussion
3.1. Final porosity
3.2. Final porosity and specific thermal energy
3.3. Porosity distribution
3.4. Microhardness
4. Conclusions.
Here is a list of suggestions for improving the manuscript:
1) Section “3. Results and Discussion” is divided into 4 subsections; but, please note that 3 of them are dedicated to “porosity”. Perhaps, the titles of these 3 subsections could be rearranged. All results are final. Why do the authors use “Final porosity”? Microhardness is also final.
2) The Abstract needs to be improved in terms of English style/phrasing; namely:
In line 21: Do not use the word “experiences”; replace it by the word “experiment”, “trial” or “test”.
(In English, “experience” means the first person effects or influence of an event or subject gained through involvement in or exposure to it). Synonyms for experience are: knowledge; involvement; skill; background; know-how; maturity; participation; patience.
In lines 25-26: Note that the phrase: … “and also higher pressures and the consolidation of compacts with lower aspects ratios” is unclear (with no verb).
3) In line 71: What do you mean by “side” parameters? It is not clear to the common readers.
4) In lines 75-76: Use “Metallographic analysis of the samples processed by this route has been carried out” OR “Metallographic analyses of the samples processed by this route have been carried out”.
5) In line 93: Instead of [4], It should be Lenel [3].
6) In line 95: I would recommend to write “of heavy metal tungsten alloy (…”.
7) In lines 96-97: I suggest the following phrasing: … and a cylindrical bar made of a temperature-resistant Cu-alloy (98.9wt% Cu, 1wt% Cr and 0.1wt% Zr).”
8) In line 103: … a suspension of graphite in acetone.
9) Note that you refer (see lines 107-108) that: “The last step consists in a cooling and pressing stage for 300 ms, when again only pressure is applied.”. What are the values of pressure applied in this last step? Note that it is important that readers are informed about all experimental details so that they can repeat the experiments if they are in possession of identical materials and equipment.
10) In line 109: replace “experience” by “experiment”.
11) In line 115: replace “experiences” by “experiments”.
12) See line 147. In English, there is a difference between section and quadrant. Section is a cutting i.e. a part cut out from the rest of something, while quadrant is one of the four sections made by dividing an area with two perpendicular lines. Is this what you intend to mean when you write “… measured in a diametrical-section quadrant in five different points, as shown in Figure 4 …)?
13) In line 164: replace “experiences” by “experiments”.
14) In line 176, refer: “heating time (t)”. And in Equation (1) use the Latin letter t instead of Greek tau.
15) In line 181: replace “experiences” by “experiments”. And in Table 2 (line 184), replace “experiences” by “experiments”.
16) In line 197, please note that the term “points cloud” is not common English terminology. Replace “points cloud” by “data points” or “scatter plot”.
17) In line 203, perhaps it is better to delete the word “conditions” …
18) In line 208, replace “points cloud” by “data points” or “scatter plot”.
19) In lines 221-222, reconsider the phrasing: “… therefore their importance”.
20) In line 292, replace “points cloud” by “data points” or “scatter plot”.
Author Response
Dear reviewer, thank you for detailed review and your interest in improving the content of this manuscript. Please find next our answer to your different comments. We hope the changes included in the manuscript are in accordance with your suggestions.
1) Section “3. Results and Discussion” is divided into 4 subsections; but, please note that 3 of them are dedicated to “porosity”. Perhaps, the titles of these 3 subsections could be rearranged. All results are final. Why do the authors use “Final porosity”? Microhardness is also final.
According to the reviewer suggestion we have modifies the titles to:
- Effect of processing parameters on porosity.
- Relationship between specific thermal energy and porosity.
- Porosity distribution inside the compacts.
Regarding the use of FINAL porosity, we just wanted to state that we are measuring the porosity in the final situation after consolidation. Based on the position of the upper electrode at any time during the sintering process, the porosity can also be known at any time of the process. The microhardness, being also final, can only be measured after the process finishes.
Taking into account your indication, we have eliminated the use of final (in this manuscript we are indeed not using any value of porosity at an intermediate situation). Only a reference is now appearing in section 2.2 to clarify this:
“On the other hand, the global porosity after the MF-ERS experiments (i.e., in the final situation after finishing the consolidation process) was calculated from…”
2) The Abstract needs to be improved in terms of English style/phrasing; namely:
In line 21: Do not use the word “experiences”; replace it by the word “experiment”, “trial” or “test”.
(In English, “experience” means the first person effects or influence of an event or subject gained through involvement in or exposure to it). Synonyms for experience are: knowledge; involvement; skill; background; know-how; maturity; participation; patience.
In lines 25-26: Note that the phrase: … “and also higher pressures and the consolidation of compacts with lower aspects ratios” is unclear (with no verb).
A proof-read has been performed in the manuscript, and changes have been included in the new version. The sentences you comment have been corrected.
3) In line 71: What do you mean by “side” parameters? It is not clear to the common readers.
We have changed to “secondary”. We hope it will be clearer.
4) In lines 75-76: Use “Metallographic analysis of the samples processed by this route has been carried out” OR “Metallographic analyses of the samples processed by this route have been carried out”.
Changed to plural.
5) In line 93: Instead of [4], It should be Lenel [3].
Right, sorry for the mistake.
6) In line 95: I would recommend to write “of heavy metal tungsten alloy (…”.
Added.
7) In lines 96-97: I suggest the following phrasing: … and a cylindrical bar made of a temperature-resistant Cu-alloy (98.9wt% Cu, 1wt% Cr and 0.1wt% Zr).”
That is clearer, we have changed the sentence.
8) In line 103: … a suspension of graphite in acetone.
Changed
9) Note that you refer (see lines 107-108) that: “The last step consists in a cooling and pressing stage for 300 ms, when again only pressure is applied.”. What are the values of pressure applied in this last step? Note that it is important that readers are informed about all experimental details so that they can repeat the experiments if they are in possession of identical materials and equipment.
We have clarified in this new version that the applied pressure in this last step is again the same as in the first step, i.e., 100 or 150 MPa.
10) In line 109: replace “experience” by “experiment”.
Done
11) In line 115: replace “experiences” by “experiments”.
Done.
12) See line 147. In English, there is a difference between section and quadrant. Section is a cutting i.e. a part cut out from the rest of something, while quadrant is one of the four sections made by dividing an area with two perpendicular lines. Is this what you intend to mean when you write “… measured in a diametrical-section quadrant in five different points, as shown in Figure 4 …)?
We think the text is right. We cut the specimen in two halves containing the diameter of the sample (a diametrical-section), and we then measure in one of the four parts appearing when the diametrical-section is divided by two perpendicular lines (a quadrant of the plane appearing after cutting). Note that the five points marked in Fig. 4 are in one of the quadrants.
13) In line 164: replace “experiences” by “experiments”.
Done
14) In line 176, refer: “heating time (t)”. And in Equation (1) use the Latin letter t instead of Greek tau.
We have changed to the Latin letter t.
15) In line 181: replace “experiences” by “experiments”. And in Table 2 (line 184), replace “experiences” by “experiments”.
Done.
16) In line 197, please note that the term “points cloud” is not common English terminology. Replace “points cloud” by “data points” or “scatter plot”.
We have changed to scatter plot.
17) In line 203, perhaps it is better to delete the word “conditions” …
The word has been deleted.
18) In line 208, replace “points cloud” by “data points” or “scatter plot”.
We have changed to scatter plot.
19) In lines 221-222, reconsider the phrasing: “… therefore their importance”.
We have rephrased the whole paragraph to make it more easily understood. This includes eliminating the commented words.
20) In line 292, replace “points cloud” by “data points” or “scatter plot”.
We have changed to data points.

Reviewer 4 Report
The article presents interesting results and describes an application regarding the electrical resistance sintering technique applied to the iron powder. However, I strongly recommend major revisions. In detail, you can find my specific points, suggested to improve the article:
- The content is clear. However, I suggest a moderate revision of the grammar. Check also the sentences, minor errors can be find, e.g. line 14 "...consolidated be the Medium..."
- Line 13: Microstructures are not present in this work. The author showed only macrographs (Figure 8). I strongly suggest to take care about the "micro" term, and delete it if microstructures are not investigated;
- The introduction is quite exhaustive and well-explains the context of the work. However, some historical description should be better described, focusing on the process characteristics and main improvements of ERS during time.
- Line 68: On the referenced bibliography, I must add that this process was deeply studied also from other groups of research. Other authors should be mentioned, e.g. Fais et al., Cannella et al., Forno et al. Here, some papers which should be briefly commented and properly referenced:
Fais, A., 2010, Processing Characteristics and Parameters in Capacitor Discharge Sintering. Journal of Materials Processing Technology, 210:22232230.http://dx.doi.org/10.1016/j.jmatprotec.2010.08.009.
Forno, I., Actis Grande, M., Fais, A., 2015, On the Application of Electro-SinterForging to the Sintering of High-Karatage Gold Powders. Gold Bulletin,48:127133. http://dx.doi.org/10.1007/s13404-015-0169-x
Fais, A., 2018, A Faster FAST: Electro-Sinter-Forging. Metal Powder Report,73:8086. http://dx.doi.org/10.1016/j.mprp.2017.06.001
Cannella, E., Nielsen, C.V., Bay, N., 2019, Process Investigation and Mechanical Properties of Electro Sinter Forged (ESF) Titanium Discs. The InternationalJournal of Advanced Manufacturing Technology (2019), 104:19851998. http://dx.doi.org/10.1007/s00170-019-03972-z.
Cannella E, Nielsen CV, Bay N (2018) Process parameter influence on electro-sinter-forging (ESF) of titanium discs. In: 18th Int. Conf.EUr. Soc. Precis. Eng. Nanotechnology, EUSPEN 2018. euspen, pp. 315-316
Cannella E et al., Resistance sintering of NdFeBCo permanent magnets and analysis of their properties, CIRP Journal of Manufacturing Science and Technologies (2020), https://doi.org/10.1016/j.cirpj.2020.02.005
- Line 72: Literature already dealt with this approach and material: see
Fais A, Actis Grande M, Forno I (2016) Influence of processingparameters on the mechanical properties of electro-sinter-forgediron based powders. Mater Des 93:458466. https://doi.org/10.1016/j.matdes.2015.12.142 - Line 89-90: How process parameters are measured? Rogowski coil for the electrical current? Load cell for the pressure? Voltmeter for voltage? or something else? please, be more specific on the used instruments
- Figure 1: Would it be possible to have a picture showing the real tool system?
- Line 104: "shaken", how can you demonstrate the repeatability of the process by shaking? Are you considering any quantitative parameter?
- Line 140: "microscopically", they are "macro"scopically analysed. No microstructures are shown. The resolution of 3 mm is not suitable for microstructural characterisation.
- Line 143: "from the dimension" Which method was used to estimate the dimensions? What is the accuracy? Why did you not use the standard Archimedes' method?
- Line 145: "may have a 5% uncertainty" How did you achieve this conclusion? the method used for the average sample porosity is very unclear and has to be better defined
- Line 146: "Vickers microhardness"What lens is used? it also misses the reference to the standard. Please refer to the ISO. 2018, BS EN 6507-1:2018 - Metallic Materials Vickers Hardness Test Part 1: Test Method. https://www.iso.org/obp/ui/#iso:std:iso:6507:-1:ed-4:v1:en
- Figure 5: I would suggest to make larger the font size , in order to make it more readable. Please, follow the same suggestion also for the other pictures/diagrams
- Line 194-195: " lower final porosities are achieved for 150 MPa just because the initial porosity is lower" I agree with that, but have you taken into account the optimal starting porosity of the compact? If so, why do not operate pressurless?
- Lines 213-228: this paragraph is very unclear. I have understood that 12 mm samples resulted in larger thermal leaks, but it should be rephrased and better explained. Avoid too long sentences: make them short and concise.
- Line 255: "the densification process is more difficult when the applied pressure is lower" Previously (lines 194-195), you stated that with low pressure you have higher heating and lower final porosities. Could you comment on this based on your previous statement?
- Line 318: Conclusion. No future works are described, at least here. What can this process enhance the actual technologies in sintering? Perhaps, move lines 315-316 into conclusions and explain in detail
- Lines 326-327: What about the difficulties you encountered with 100 MPa in increasing the density of the compact (Line 255)?
Author Response
Dear reviewer, thank you for detailed review and your interest in improving the content of this manuscript. Please find next our answer to your different comments. We hope the changes included in the manuscript are in accordance with your suggestions.
- The content is clear. However, I suggest a moderate revision of the grammar. Check also the sentences, minor errors can be find, e.g. line 14"...consolidated be the Medium..."
A proof-read has been performed in the manuscript, and changes have been included in the new version.
- Line 13: Microstructures are not present in this work. The author showed only macrographs (Figure 8). I strongly suggest to take care about the "micro" term, and delete it if microstructures are not investigated;
We have corrected the manuscript, avoiding the use of the term microstructure.
- The introduction is quite exhaustive and well-explains the context of the work. However, some historical description should be better described, focusing on the process characteristics and main improvements of ERS during time.
Regarding this point, we have a problem, because one of the reviewers asks to make the opposite (“The first three paragraphs of the introduction are about the historical background of sintering metallic powders. I think it can be concisely summarized into a single paragraph since it is not very important.”). The only option we find is to maintain the same content as was initially. Sorry about this.
- Line 68: On the referenced bibliography, I must add that this process was deeply studied also from other groups of research. Other authors should be mentioned, e.g. Fais et al., Cannella et al., Forno et al. Here, some papers which should be briefly commented and properly referenced:
Fais, A., 2010, Processing Characteristics and Parameters in Capacitor Discharge Sintering. Journal of Materials Processing Technology, 210:22232230. http://dx.doi.org/10.1016/j.jmatprotec.2010.08.009.
Forno, I., Actis Grande, M., Fais, A., 2015, On the Application of Electro-Sinter Forging to the Sintering of High-Karatage Gold Powders. Gold Bulletin,48:127133. http://dx.doi.org/10.1007/s13404-015-0169-x
Fais, A., 2018, A Faster FAST: Electro-Sinter-Forging. Metal Powder Report,73:8086. http://dx.doi.org/10.1016/j.mprp.2017.06.001
Cannella, E., Nielsen, C.V., Bay, N., 2019, Process Investigation and Mechanical Properties of Electro Sinter Forged (ESF) Titanium Discs. The International Journal of Advanced Manufacturing Technology (2019), 104:19851998. http://dx.doi.org/10.1007/s00170-019-03972-z.
Cannella E, Nielsen CV, Bay N (2018) Process parameter influence on electro-sinter-forging (ESF) of titanium discs. In: 18th Int. Conf.EUr. Soc. Precis. Eng. Nanotechnology, EUSPEN 2018. euspen, pp. 315-316
Cannella E et al., Resistance sintering of NdFeBCo permanent magnets and analysis of their properties, CIRP Journal of Manufacturing Science and Technologies (2020), https://doi.org/10.1016/j.cirpj.2020.02.005
We have included in the introduction section references to the other groups working with this process, and a brief reference to the nature of the studied materials.
- Line 72: Literature already dealt with this approach and material: see
Fais A, Actis Grande M, Forno I (2016) Influence of processing parameters on the mechanical properties of electro-sinter-forged iron based powders. Mater Des 93:458466. https://doi.org/10.1016/j.matdes.2015.12.142
This reference has been included in the Introduction section.
- Line 89-90: How process parameters are measured? Rogowski coil for the electrical current? Load cell for the pressure? Voltmeter for voltage? or something else? please, be more specific on the used instruments
As mentioned in section 2.1, the equipment used is a commercial resistance welding equipment adapted for this process. We have not access to the electronic control of this equipment, and cannot exactly indicate the internal instruments that provide the electronic data. The instrument manual only indicates that both the current and voltage are not measured in the primary, but in the secondary circuit, therefore decreasing the measuring error. In particular, there are two different systems to measure the current intensity, by a coil and by the Shunt effect. We however have not a precise knowledge about this and prefer not to include it in the manuscript. The company (Serra Soldadura) is mentioned in the manuscript and we prefer readers to ask directly for these details to the company in case they have a specific interest on this.
- Figure 1: Would it be possible to have a picture showing the real tool system?
Fig 1 now includes the real die used in the experiments. We have however included a recreation of the inner part to maintain the information given in the previous version of this Figure.
- Line 104: "shaken", how can you demonstrate the repeatability of the process by shaking? Are you considering any quantitative parameter?
Yes, we follow a standard to reach the tap porosity of the powder, as indicate in the text. This is always the initial condition before starting any of the sintering process. Please see ref [20].
- Line 140: "microscopically", they are "macro"scopically analysed. No microstructures are shown. The resolution of 3 mm is not suitable for microstructural characterisation.
According to this indication we have changed in the text to “macroscopically”.
- Line 143: "from the dimension" Which method was used to estimate the dimensions? What is the accuracy? Why did you not use the standard Archimedes' method?
Dimensions were determined by measuring the height and diameter of the specimen with a micrometer (in eight positions around the specimen and computing the mean values). The instrument accuracy is 10 microns. Regarding weigh the accuracy is 0.1 mg.
The main source of error is the mean value obtained from the two measured specimens, more than that coming from the instruments used.
Regarding Archimedes, we have the problem of having specimens with a relatively high porosity in the external layer. This makes the porosity to be interconnected and the fluid can gets inside the specimen, giving a wrong value. The solution is using a layer covering this external porosity, but our experience says that results are not very accurate with this technique.
- Line 145: "may have a 5% uncertainty" How did you achieve this conclusion? the method used for the average sample porosity is very unclear and has to be better defined
The 5% mainly comes from the measurement of the compact volume. Unfortunately, despite taking care in the processes, the obtained compacts do not have a perfect uniform height (it is difficult to guarantee the exact wafers position), neither a perfect circular section (the ceramic die has not an exact circular shape). In particular, the value partially comes from comparing the results obtained with the usual measuring procedure (with a micrometer) and those coming from experiments carried out with a much more precise instrument (laser profilometer). On the other hand, we also compared the difference in the dimension of different samples processed under the same condition. The conjunction study of this two uncertainty sources results in the maximum value of 5%.
We have included in the text of this new version the way we have determined the dimensions of the specimen (diameter and height measured at eight different position for each specimen).
- Line 146: "Vickers microhardness" What lens is used? it also misses the reference to the standard. Please refer to the ISO. 2018, BS EN 6507-1:2018 - Metallic Materials Vickers Hardness Test Part 1: Test Method. https://www.iso.org/obp/ui/#iso:std:iso:6507:-1:ed-4:v1:en
We are using an automatic microhardness tester (EMCO tester), and unfortunately have not the data of the lens used for a particular image automatically captured, shown and measured in the equipment.
The reference to the standards is now included in the text.
- Figure 5: I would suggest to make larger the font size, in order to make it more readable. Please, follow the same suggestion also for the other pictures/diagrams.
Font size has been increased in Fig 5. Regarding the rest of the figures, we find that the font size is similar to the manuscript text size. We have at the moment leaved the same size, although depending on the final size of these figures we are of course open to change them if necessary.
- Line 194-195: "lower final porosities are achieved for 150 MPa just because the initial porosity is lower" I agree with that, but have you taken into account the optimal starting porosity of the compact? If so, why do not operate pressureless?
We are limited in this technique by the load that necessarily has to apply the equipment to work, and by the maximum load of 15 kN that the equipment can apply. We cannot work without applying pressure, the voltage difference and the bad contact between particles and with electrodes would provoke arcs / sparks, or, in case of actually existing a way for the current to flow, the system resistance would be so high that current would not flow. On the other hand, higher pressures require the use of dies with lower sections, therefore taking more importance the effect on the external layers.
According to our previous experiences (values of 80, 70 and even 50 MPa have been tested in the past), an appropriate pressure for this technique to work properly is around the values we have tested in this manuscript.
- Lines 213-228: this paragraph is very unclear. I have understood that 12 mm samples resulted in larger thermal leaks, but it should be rephrased and better explained. Avoid too long sentences: make them short and concise.
We have tried to clarify the explanation. The content is the same, but we think that the reader can now better follow the discourse.
- Line 255: "the densification process is more difficult when the applied pressure is lower" Previously (lines 194-195), you stated that with low pressure you have higher heating and lower final porosities. Could you comment on this based on your previous statement?
Please, note that in lines 194-195 we stated that with 150 MPa the final porosity is lower (just because the initial porosity is lower). Indeed the densification process (measured as the change of porosity with the process) is some more efficient for low pressure, because the resistivity is higher and the thermal energy released is higher.
Now, in line 255, when comparing the different macrographs, we said that the densification is more difficult for 100 MPa, just because the final porosity is lower and images show a higher area with more porosity.
We agree that this is confusing. According to the reviewer comment, we prefer to change the comment in line 255, and talk about the final porosity instead of the densification process.
- Line 318: Conclusion. No future works are described, at least here. What can this process enhance the actual technologies in sintering? Perhaps, move lines 315-316 into conclusions and explain in detail.
Acoording to this comment, we have added a final paragraph in the Conclusions section regarding the use of numerical simulation techniques to improve the knowledge and results of this technique.
- Lines 326-327: What about the difficulties you encountered with 100 MPa in increasing the density of the compact (Line 255)?
We consider it is indeed a problem of the studied conditions. It is possible to increase the density for 100 or for 150 MPa, but more extreme conditions of intensity or time have to be used. In order to make this, the weld of the compact with the wafers has to be avoided. We have included in the text this idea at the end of Section 3.3, indicating that the use of improved non-stick alloys for the wafers, or the use of graphite deposited on the wafers surface, have to be considered. Also, a connection with the simulation processes we carry out with this technique is commented in the last paragraph of section 3.4.

Round 2
Reviewer 1 Report
I am pleased with the improvement in the content of this manuscript.
Author Response
Ok,
Many thanks.
Reviewer 2 Report
Thank you for the revisions. There is significant improvement in the manuscript, especially the tables and figures. However, there are still grammar mistakes in the manuscript, so please perform another proof-read. Additionally, two reviewers commented on the need for the improvement of the introduction but the authors replied “The only option we find is to maintain the same content as was initially”. This is clearly unacceptable and there must be some changes made.
Author Response
Dear reviewer, thank you for your new revision of the manuscript. Please find next our answer to your different comments. We hope the changes included in the manuscript are in accordance with your suggestions.
Thank you for the revisions. There is significant improvement in the manuscript, especially the tables and figures. However, there are still grammar mistakes in the manuscript, so please perform another proof-read. Additionally, two reviewers commented on the need for the improvement of the introduction but the authors replied “The only option we find is to maintain the same content as was initially”. This is clearly unacceptable and there must be some changes made.
A new proof-read has been performed in the manuscript, and changes have been included in the new version. We hope with this proof-read and the final corrections by the Editorial Team the manuscript will finally be ready for publication.
Regarding the Introduction section, please note that we have made some changes according to the different reviewers suggestions, including new references and comments. We however decided not to change the three paragraphs of the historical background because one of the reviewers had the opposite opinion to yours, i.e. increasing the content of the historical background. Nevertheless we have summarized those three paragraphs in only one in this new version of the manuscript.
Reviewer 4 Report
Dear Authors. Thanks for giving your good revisions and clear answers to my points. I think the paper was improved if compared to the first version.
Author Response
Ok,
Many thanks